# RT-2: Vision-Language-Action Models Transfer Web Knowledge to Robotic Control

Anthony Brohan, Noah Brown, Justice Carbajal, Yevgen Chebotar, Xi Chen,
Krzysztof Choromanski, Tianli Ding, Danny Driess, Avinava Dubey, Chelsea Finn,
Pete Florence, Chuyuan Fu, Montse Gonzalez Arenas, Keerthana Gopalakrishnan, Kehang Han,
Karol Hausman, Alexander Herzog, Jasmine Hsu, Brian Ichter, Alex Irpan, Nikhil Joshi,
Ryan Julian, Dmitry Kalashnikov, Yuheng Kuang, Isabel Leal, Lisa Lee, Tsang-Wei Edward Lee,
Sergey Levine, Yao Lu, Henryk Michalewski, Igor Mordatch, Karl Pertsch, Kanishka Rao,
Krista Reymann, Michael Ryoo, Grecia Salazar, Pannag Sanketi, Pierre Sermanet, Jaspiar Singh,
Anikait Singh, Radu Soricut, Huong Tran, Vincent Vanhoucke, Quan Vuong, Ayzaan Wahid, Stefan Welker,
Paul Wohlhart, Jialin Wu, Fei Xia, Ted Xiao, Peng Xu, Sichun Xu, Tianhe Yu, Brianna Zitkovich

Google DeepMind *

**Abstract:** We study how vision-language models trained on Internet-scale data can be incorporated directly into end-to-end robotic control to boost generalization and enable emergent semantic reasoning. Our goal is to enable a single end-to-end trained model to both learn to map robot observations to actions and enjoy the benefits of large-scale pre-training on language and vision-language data from the web. To this end, we propose to co-fine-tune state-of-the-art vision-language models on both robotic trajectory data and Internet-scale vision-language tasks, such as visual question answering. In contrast to other approaches, we propose a simple, general recipe to achieve this goal: in order to fit both natural language responses and robotic actions into the same format, we express the actions as text tokens and incorporate them directly into the training set of the model in the same way as natural language tokens. We refer to such category of models as vision-language-action models (VLA) and instantiate an example of such a model, which we call RT-2. Our extensive evaluation (6k evaluation trials) shows that our approach leads to performant robotic policies and enables RT-2 to obtain a range of emergent capabilities from Internet-scale training. This includes significantly improved generalization to novel objects, the ability to interpret commands not present in the robot training data (such as placing an object onto a particular number or icon), and the ability to perform rudimentary reasoning in response to user commands (such as picking up the smallest or largest object, or the one closest to another object). We further show that incorporating chain of thought reasoning allows RT-2 to perform multi-stage semantic reasoning, for example figuring out which object to pick up for use as an improvised hammer (a rock), or which type of drink is best suited for someone who is tired (an energy drink).

## 1 Introduction

High-capacity models pretrained on broad web-scale datasets provide an effective and powerful platform for a wide range of downstream tasks: large language models can enable not only fluent text generation [1, 2, 3] but emergent problem-solving [4, 5, 6] and creative generation of prose [7, 2] and code [8], while vision-language models enable open-vocabulary visual recognition [9, 10, 11] and can even make complex inferences about object-agent interactions in images [12, 13, 14, 15, 16, 17, 18]. Such semantic reasoning, problem solving, and visual interpretation capabilities would be tremendously useful for generalist robots that must perform a variety of tasks in real-world environments. However, it is unclear how robots should acquire such capabilities. While a brute force approach might entail collecting millions of robotic interaction trials, the most capable language and vision-language models are trained on billions of tokens and images from the web [12, 15, 16, 18] – an amount unlikely to be matched with robot data in the near future. On the other hand, directly applying such models to robotic tasks is also difficult: such models reason about semantics, labels, and textual prompts, whereas robots require grounded low-level actions, such as Cartesian end-effector commands. While a number of recent works have sought to incorporate language models (LLMs) and vision-language models (VLMs) into robotics [19, 17, 20], such methods generally address only the "higher level" aspects of robotic planning, essentially taking the role of a state machine that interprets commands and parses them into individual primitives (such as picking and placing objects), which are then executed by separate low-level controllers that themselves do not benefit from the rich semantic knowledge of Internet-scale models during training. Therefore, in this paper we ask: can large pretrained vision-language models be integrated directly into low-level robotic control to boost generalization and enable emergent semantic reasoning?

---

* Authors listed in alphabetical order, with contributions listed in Appendix A.
Corresponding emails: `chebotar@google.com`, `tianheyu@google.com`, `karolhausman@google.com`

7th Conference on Robot Learning (CoRL 2023), Atlanta, USA.

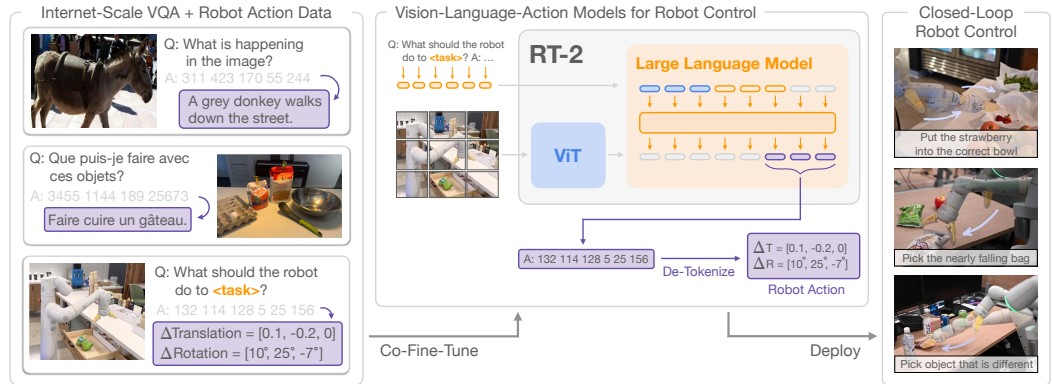

Figure 1: RT-2 overview: we represent robot actions as another language, which can be cast into text tokens and trained together with Internet-scale vision-language datasets. During inference, the text tokens are de-tokenized into robot actions, enabling closed loop control. This allows us to leverage the backbone and pretraining of vision-language models in learning robotic policies, transferring some of their generalization, semantic understanding, and reasoning to robotic control. We demonstrate examples of RT-2 execution on the project website: robotics-transformer2.github.io.

To this end, we explore an approach that is both simple and surprisingly effective: we directly train vision-language models designed for open-vocabulary visual question answering and visual dialogue to output low-level robot actions, along with solving other Internet-scale vision-language tasks. Although such models are typically trained to produce natural language tokens, we can train them on robotic trajectories by *tokenizing the actions into text tokens* and creating "multimodal sentences" [17] that "respond" to robotic instructions paired with camera observations by producing corresponding actions. In this way, vision-language models can be directly trained to act as instruction following robotic policies. This simple approach is in contrast with prior alternatives for incorporating VLMs into robot policies [21] or designing new vision-language-action architectures from scratch [22]: instead, pre-existing vision-language models, with already-amortized significant compute investment, are trained without any new parameters to output text-encoded actions. We refer to this category of models as vision-language-action (VLA) models. We instantiate VLA models by building on the protocol proposed for RT-1 [1], using a similar dataset, but expanding the model to use a large vision-language backbone. Hence we refer to our model as RT-2 (Robotics Transformer 2). We provide an overview in Figure 1.

We observe that robotic policies derived from such vision-language models exhibit a range of remarkable capabilities, combining the physical motions learned from the robot data with the ability to interpret images and text learned from web data into a single model. Besides the expected benefit of dramatically improving generalization to novel objects and semantically varied instructions, we observe a number of emergent capabilities. While the model's physical skills are still limited to the distribution of skills seen in the robot data, the model acquires the ability to deploy those skills in new ways by interpreting images and language commands using knowledge gleaned from the web. Some example highlights are shown in Figure 2. The model is able to re-purpose pick and place skills learned from robot data to place objects near semantically indicated locations, such as specific numbers or icons, despite those cues not being present in the robot data. The model can also interpret relations between objects to determine which object to pick and where to place it, despite no such relations being provided in the robot demonstrations. Furthermore, if we augment the command with chain of thought prompting, the model is able to make even more complex semantic inferences, such as figuring out which object to pick up for use as an improvised hammer (a rock), or which type of drink is best suited for someone who is tired (an energy drink).

Our main contribution is RT-2, a family of models derived from fine-tuning large vision-language models trained on web-scale data to directly act as generalizable and semantically aware robotic policies. Our experiments investigate models with up to 55B parameters trained on Internet data and instruction-annotated robotic trajectories from previous work [1]. Over the course of 6k robotic evaluations, we show that RT-2 enable significant improvements to generalization over objects, scenes, and instructions, and exhibit a breadth of emergent capabilities inherited from web-scale vision-language pretraining.

## 2 Related Work

**Vision-language models.** There are several categories of *Vision-Language Models* (VLMs) [23], with perhaps two most relevant: (1) representation-learning models, e.g. CLIP [9], which learn common embeddings for both modalities, and (2) visual language models of the form {vision,text} → {text} which learn to take vision and language as input and provide free-form text. Both categories have been used to provide pretraining for a wide variety of applied to downstream applications such as object classification [9], detec-

tion [24], and segmentation [25]. In this work, we focus on the latter category [12, 15, 16, 17, 26, 27, 13, 28]. These models are generally trained on many different tasks, such as image captioning, vision-question answering (VQA), and general language tasks on multiple datasets at the same time. While prior works study VLMs for a wide range of problems and settings including in robotics, our focus is on how the capabilities of VLMs can be extended to robotics closed-loop control by endowing them with the ability to predict robot actions, thus leveraging the knowledge already present in VLMs to enable new levels of generalization.

**Generalization in robot learning.** Developing robotic controllers that can broadly succeed in a variety of scenarios is a long-standing goal in robotics research [29, 30]. A promising approach for enabling generalization in robotic manipulation is by learning from large and diverse datasets [31, 32, 33]. By doing so, prior methods have demonstrated how robots can generalize to novel object instances [31, 34, 32, 35, 36], to tasks involving novel combinations of objects and skills [37, 38, 39, 40, 41], to new goals or language instructions [42, 43, 41, 44, 45, 46, 47], to tasks with novel semantic object categories [48, 49], and to unseen environments [50, 51, 52]. Unlike most of these prior works, we aim to develop and study a single model that can generalize to unseen conditions along all of these axes. A key ingredient of our approach is to leverage pre-trained models that have been exposed to data that is much broader than the data seen by the robot.

**Pre-training for robotic manipulation.** Pre-training has a long history in robotic learning. Most works focus on pre-trained visual representations that can be used to initialize the encoder of the robot's camera observations, either via supervised ImageNet classification [53], data augmentation [54, 55, 56, 57] or objectives that are tailored towards robotic control [58, 59, 60, 61, 62]. Other works have incorporated pre-trained language models, often either as an instruction encoder [63, 64, 43, 41, 44, 1, 65] or for high-level planning [66, 19, 17, 67, 68, 69]. Rather than using pre-training vision models or pre-trained language models, we specifically consider the use of pre-trained vision-language models (VLMs), which provide rich, grounded knowledge about the world. Prior works have studied the use of VLMs for robotics [48, 61, 49, 17, 70, 71, 72], and form part of the inspiration for this work. These prior approaches use VLMs for visual state representations [61], for identifying objects [49, 70], for high-level planning [17], or for providing supervision or success detection [73, 72, 74, 75, 76]. While CLIPort [48] and MOO [49] integrate pre-trained VLMs into end-to-end visuomotor manipulation policies, both incorporate significant structure into the policy that limits their applicability. Notably, our work does not rely on a restricted 2D action space and does not require a calibrated camera. Moreover, a critical distinction is that, unlike these works, we leverage VLMs that generate language, and the unified output space of our formulation enables model weights to be entirely shared across language and action tasks, without introducing action-only model layer components.

## 3 Vision-Language-Action Models

In this section, we present our model family and the design choices for enabling training VLMs to directly perform closed-loop robot control. First, we describe the general architecture of our models and how they can be derived from models that are commonly used for vision-language tasks. Then, we introduce the recipe and challenges of fine-tuning large VLMs that are pre-trained on web-scale data to directly output robot actions, becoming VLA models. Finally, we describe how to make these models practical for robot tasks, addressing challenges with model size and inference speed to enable real-time control.

### 3.1 Pre-Trained Vision-Language Models

The vision-language models [16, 17] that we build on in this work take as input one or more images and produce a sequence of tokens, which conventionally represents natural language text. Such models can perform a wide range of visual interpretation and reasoning tasks, from inferring the composition of an image to answering questions about individual objects and their relations to other objects [12, 16, 17, 18]. Representing the knowledge necessary to perform such a wide range of tasks requires large models and web-scale datasets. In this work, we adapt two previously proposed VLMs to act as VLA models: PaLI-X [16] and PaLM-E [17]. We will refer to vision-language-action versions of these models as RT-2-PaLI-X and RT-2-PaLM-E. We leverage instantiations of these models that range in size from billions to tens of billions of parameters. We provide a detailed description of the architecture of these two models in Appendix D.

### 3.2 Robot-Action Fine-tuning

To enable vision-language models to control a robot, they must be trained to output actions. We take a direct approach to this problem, representing actions as tokens in the model's output, which are treated in the same way as language tokens. We base our action encoding on the discretization proposed by Brohan et al. [1] for the RT-1 model. The action space consists of 6-DoF positional and rotational displacement of the robot end-effector, as well as the level of extension of the robot gripper and a special discrete command for terminating the episode, which should be triggered by the policy to signal successful completion. The continuous dimensions (all dimensions except for the discrete termination command) are discretized into 256 bins uniformly. Thus, the robot action can be represented using ordinals of the

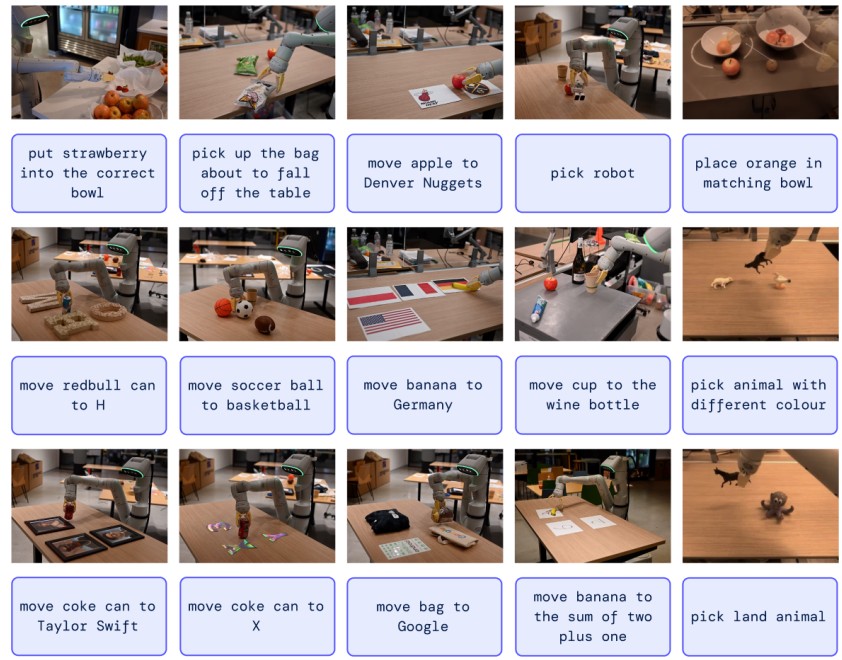

Figure 2: RT-2 is able to generalize to a variety of real-world situations that require reasoning, symbol understanding, and human recognition. We study these challenging scenarios in detail in Section 4.

discrete bins as 8 integer numbers. In order to use these discretized actions to finetune a vision-language into a vision-language-*action* model, we need to associate tokens from the model's *existing* tokenization with the discrete action bins. This requires reserving 256 tokens to serve as action tokens. Which tokens to choose depends on the particular tokenization used by each VLM, which we discuss later in this section. In order to define a target for VLM fine-tuning we convert the action vector into a single string by simply concatenating action tokens for each dimension with a space character:

$$\text{"terminate } \Delta\text{pos}_x \ \Delta\text{pos}_y \ \Delta\text{pos}_z \ \Delta\text{rot}_x \ \Delta\text{rot}_y \ \Delta\text{rot}_z \ \text{gripper\_extension"}.$$

A possible instantiation of such a target could be: "1 128 91 241 5 101 127". The two VLMs that we finetune in our experiments, PaLI-X [16] and PaLM-E [17], use different tokenizations. For PaLI-X, integers up to 1000 each have a unique token, so we simply associate the action bins to the token representing the corresponding integer. For the PaLM-E model, which does not provide this convenient representation of numbers, we simply overwrite the 256 least frequently used tokens to represent the action vocabulary. It is worth noting that training VLMs to override existing tokens with action tokens is a form of symbol tuning [77], which has been shown to work well for VLMs in prior work.

Taking the action representation described above, we convert our robot data to be suitable for VLM model fine-tuning, where our inputs include robot camera image and textual task description (using standard VQA format "Q: what action should the robot take to [task instruction]? A:"), and our output is formatted as a string of numbers/least frequently used tokens representing a robot action.

**Co-Fine-Tuning.** As we will show in our experiments, a key technical detail of the training recipe that improves robot performance is *co-fine-tuning* robotics data with the original web data instead of naïve finetuning on robot data only. We notice that co-fine-tuning leads to more generalizable policies since the policies are exposed to both abstract visual concepts from web scale data and low level robot actions during fine-tuning, instead of just robot actions. During co-fine-tuning we balance the ratios of robot and web data in each training batch by increasing the sampling weight on the robot dataset.

**Output Constraint.** One important distinction between RT-2 and standard VLMs is that RT-2 is required to output valid action tokens for execution on the real robot. Thus, to ensure that RT-2 outputs valid action tokens during decoding, we constrain its output vocabulary via only sampling valid action tokens when the model is prompted with a robot-action task, whereas the model is still allowed to output the full range of natural language tokens on standard vision-language tasks.

### 3.3 Real-Time Inference

The size of modern VLMs can reach tens or hundreds of billions of parameters [16, 17]. The largest model trained in this work uses 55B parameters. It is infeasible to directly run such models on the standard

desktop-style machines or on-robot GPUs commonly used for real-time robot control. To the best of our knowledge, our model is the largest ever, by over an order of magnitude, used for direct closed-loop robotic control, and therefore requires a new set of solutions to enable efficient real-time inference. We develop a protocol that allows us to run RT-2 models on robots by deploying them in a multi-TPU cloud service and querying this service over the network. With this solution, we can achieve a suitable frequency of control and also serve multiple robots using the same cloud service. The largest model we evaluated, the 55B parameter RT-2-PaLI-X-55B model, can run at a frequency of 1-3 Hz. The smaller version of that model, consisting of 5B parameters, can run at a frequency of around 5 Hz.

## 4  Experiments

Our experiments focus on real-world generalization and emergent capabilities of RT-2 and aim to answer the following questions:

1. How does RT-2 perform on seen tasks and more importantly, generalize over new objects, backgrounds, and environments?
2. Can we observe and measure any emergent capabilities of RT-2?
3. How does the generalization vary with parameter count and other design decisions?
4. Can RT-2 exhibit signs of chain-of-thought reasoning similarly to vision-language models?

We evaluate our approach and several baselines with about 6,000 evaluation trajectories in a variety of conditions, which we describe in the following sections. Unless specified otherwise, we use a 7DoF mobile manipulator with the action space described in Sec. 3.2. We also demonstrate examples of RT-2 execution on the project website: `robotics-transformer2.github.io`. We train two specific instantiations of RT-2 that leverage pre-trained VLMs: (1) **RT-2-PaLI-X** is built from 5B and 55B PaLI-X [16], and (2) **RT-2-PaLM-E** is built from 12B PaLM-E [17].

For training, we leverage the original web scale data from Chen et al. [16] and Driess et al. [17], which consists of visual question answering, captioning, and unstructured interwoven image and text examples. We combine it with the robot demonstration data from Brohan et al. [1], which was collected with 13 robots over 17 months in an office kitchen environment. Each robot demonstration trajectory is annotated with a natural language instruction that describes the task performed, consisting of a verb describing the skill (e.g., "pick", "open", "place into") and one or more nouns describing the objects manipulated (e.g., "7up can", "drawer", "napkin") (see Appendix B for more details on the used datasets). For all RT-2 training runs we adopt the hyperparameters from the original PaLI-X [16] and PaLM-E [17] papers, including learning rate schedules and regularizations. More training details can be found in Appendix E.

**Baselines.** We compare our method to multiple state-of-the-art baselines that challenge different aspects of our method. All of the baselines use the exact same robotic data. To compare against a state-of-the-art policy, we use **RT-1** [1], a 35M parameter transformer-based model. To compare against state-of-the-art pretrained representations, we use **VC-1** [78] and **R3M** [58], with policies implemented by training an RT-1 backbone to take their representations as input. To compare against other architectures for using VLMs, we use **MOO** [49], which uses a VLM to create an additional image channel for a semantic map, which is then fed into an RT-1 backbone. More information is provided in Appendix C.

### 4.1  How does RT-2 perform on seen tasks and more importantly, generalize over new objects, backgrounds, and environments?

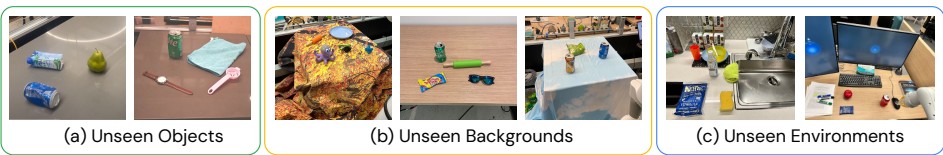

(a) Unseen Objects          (b) Unseen Backgrounds          (c) Unseen Environments

Figure 3: Example generalization scenarios used for evaluation in Figures 4 and 6b and Tables 5 and 7.

To evaluate in-distribution performance as well as generalization capabilities, we compare the RT-2-PaLI-X and RT-2-PaLM-E models to the four baselines listed in the previous sections. For the *seen tasks* category, we use the same suite of seen instructions as in RT-1 [1], which include over 200 tasks in this evaluation: 36 for picking objects, 35 for knocking objects, 35 for placing things upright, 48 for moving objects, 18 for opening and closing various drawers, and 36 for picking out of and placing objects into drawers. Note, however, that these "in-distribution" evaluations still vary the placement of objects and factors such as time of day and robot position, requiring the skills to generalize to realistic variability in the environment.

Figure 3 shows example generalization evaluations, which are split into *unseen* categories (*objects*, *backgrounds* and *environments*), and are additionally split into easy and hard cases. For unseen objects, hard

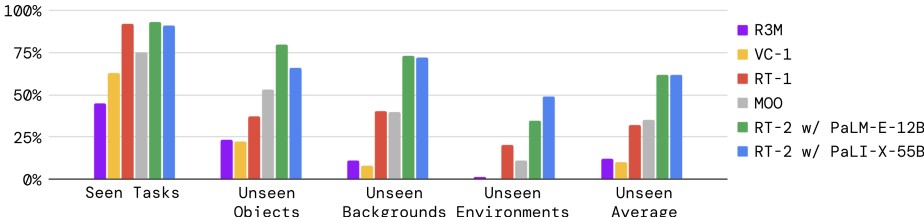

Figure 4: Overall performance of two instantiations of RT-2 and baselines across seen training tasks as well as unseen evaluations measuring generalization to novel objects, novel backgrounds, and novel environments. Appendix Table 5 details the full results.

cases include harder-to-grasp and more unique objects (such as toys). For unseen backgrounds, hard cases include more varied backgrounds and novel objects. Lastly, for unseen environments, hard cases correspond to a more visually distinct office desk environment with monitors and accessories, while the easier environment is a kitchen sink. These evaluations consists of over 280 tasks that focus primarily on pick and placing skills in many diverse scenarios. The list of instructions for unseen categories is specified in Appendix F.3.

The evaluation results are shown in Figure 4 and Appendix Table 5. The performance on seen tasks is similar between the RT-2 models and RT-1, with other baselines attaining a lower success rate. The difference between the RT-2 models and the baseline is most pronounced in the various generalization experiments, suggesting that the strength of vision-language-action models lies in transferring more generalizable visual and semantic concepts from their Internet-scale pretraining data. Here, on average, both instantiations of RT-2 perform similarly, resulting in ~2x improvement over the next two baselines, RT-1 and MOO, and ~6x better than the other baselines. The PaLM-E version of RT-2 seems to perform better than the RT-2-PaLI-X in harder versions of generalization scenarios while under-performing on easier ones, resulting in a similar average performance.

**Open Source Language Table Benchmark.** To provide an additional point of comparison using open-source baselines and environments, we leverage the open-source Language-Table simulation environment from Lynch et al. [79]. We co-fine-tune a smaller PaLI 3B model on several prediction tasks, including in-domain VQA tasks, for the Language-Table dataset, and evaluate the resulting policy in simulation. For the action prediction task, we discretize and encode actions as text in the format "X Y", where X and Y range between{-10, -9, ..., +9, +10}, and represent delta 2D cartesian setpoints of the end effector. Due to its reduced size, the resulting model can run inference at a similar rate (5 Hz) as the other baselines. The results of this experiment are presented in Table 1. We observe a significant performance boost when using our model compared to the baselines, indicating that the VLM-based pre-training together with the expressiveness of the large PaLI model can be beneficial in other scenarios, in this case, simulation with a different robot. We also show qualitative real-world out-of-distribution behaviors behaviors in Figure 5, demonstrating novel pushing tasks and targeting objects not before seen in this environment. More details about the Language Table experiments can be found in Appendix B and D.

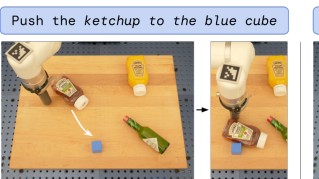 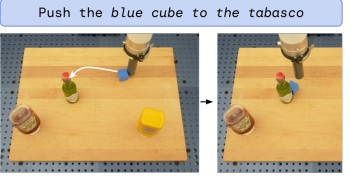

Figure 5: Real-world out-of-distribution behaviors in the Language Table environment. Identical RT-2-PaLI-3B model checkpoint is used as in Tab. 1.

Table 1: Performance on the simulated Language-Table tasks [64].

| Model | Language-Table |
|---|---|
| BC-Zero [41] | $72 \pm 3$ |
| RT-1 [1] | $74 \pm 13$ |
| LAVA [79] | $77 \pm 4$ |
| **RT-2-PaLI-3B (ours)** | **$90 \pm 10$** |

### 4.2 Can we observe and measure any emergent capabilities of RT-2?

In addition to evaluating the generalization capabilities of vision-language-action models, we also aim to evaluate the degree to which such models can enable new capabilities beyond those demonstrated in the robot data by transferring knowledge from the web. We refer to such capabilities as *emergent*, in the sense that they emerge by transferring Internet-scale pretraining. We do not expect such transfer to enable new robotic *motions*, but we do expect semantic and visual concepts, including relations and nouns, to transfer effectively, even in cases where those concepts were not seen in the robot data.

**Qualitative Evaluations.** First, we experiment with our RT-2-PaLI-X model to determine various emergent capabilities transferred from vision-language concepts. We demonstrate some examples of such interactions in Figure 2. We find through our explorations that RT-2 inherits novel capabilities in terms of semantic understanding and basic reasoning in the context of the scene. For example accomplishing the task "put strawberry into the correct bowl" requires a nuanced understanding of not only what a strawberry and bowl are, but also

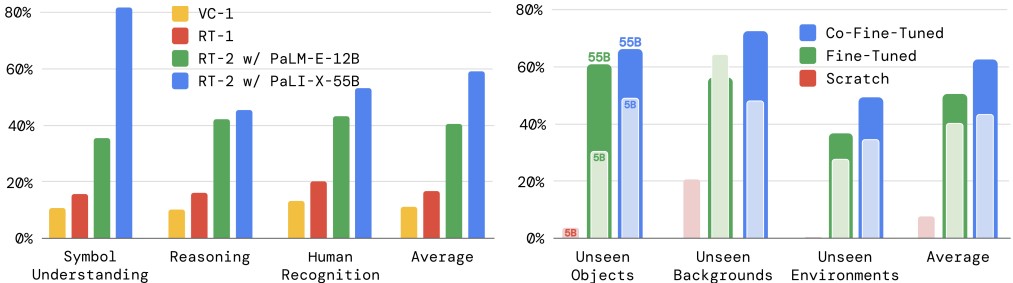

(a) Performance comparison on various emergent skill evaluations (Figure 8) between RT-2 and two baselines.

(b) Ablations of RT-2-PaLI-X showcasing the impact of parameter count and training strategy on generalization.

Figure 6: Quantitative performance of RT-2 across (6a) emergent skills and (6b) size and training ablations. Appendix Tables 6 and 7 detail the full numerical results.

reasoning in the context the scene to know the strawberry should go with the like fruits. For the task "pick up the bag about to fall off the table," RT-2 demonstrates physical understanding to disambiguate between two bags and recognize the precariously placed object. All the interactions tested in these scenarios have never been seen in the robot data, which points to the transfer of semantic knowledge from vision-language data.

**Quantitative Evaluations.** To quantify these emergent capabilities, we take the top two baselines from the previous evaluations, RT-1 and VC-1, and compare them against our two models: RT-2-PaLI-X and RT-2-PaLM-E. To reduce the variance of these experiment, we evaluate all of the methods using the A/B testing framework [80], where all four models are evaluated one after another in the exact same conditions.

We' split the emergent capabilities of RT-2 into three categories covering axes of reasoning and semantic understanding (with examples of each shown in Appendix Figure 8). The first we term *symbol understanding*, which explicitly tests whether the RT-2 policy transfers semantic knowledge from vision-language pretraining that was not present in any of the robot data. Example instructions in this category are "move apple to 3" or "push coke can on top of heart". The second category we term *reasoning*, which demonstrates the ability to apply various aspects of reasoning of the underlying VLM to control tasks. These tasks require visual reasoning ("move the apple to cup with same color"), math ("move X near the sum of two plus one"), and multilingual understanding ("mueve la manzana al vaso verde"). We refer to the last category as *human recognition* tasks, which include tasks such as "move the coke can to the person with glasses", to demonstrate human-centric understanding and recognition. The full list of instructions used for this evaluation is specified in Appendix F.3.

We present the results of this experiment in Figure 6a with all the numerical results in Appendix H.2. We observe that our VLA models significantly outperform the baselines across all categories, with our best RT-2-PaLI-X model achieving more than 3x average success rate over the next best baseline (RT-1). We also note that while the larger PaLI-X-based model results in better symbol understanding, reasoning and person recognition performance on average, the smaller PaLM-E-based model has an edge on tasks that involve math reasoning. We attribute this interesting result to the different pre-training mixture used in PaLM-E, which results in a model that is more capable at math calculation than the mostly visually pre-trained PaLI-X.

### 4.3 How does the generalization vary with parameter count and other design decisions?

For this comparison, we use RT-2-PaLI-X model because of its flexibility in terms of the model size (due to the nature of PaLM-E, RT-2-PaLM-E is restricted to only certain sizes of PaLM and ViT models). In particular, we compare two different model sizes, 5B and 55B, as well as three different training routines: training a model from scratch, without using any weights from the VLM pre-training; fine-tuning a pre-trained model using robot action data only; and co-fine-tuning (co-training with fine-tuning), the primary method used in this work where we use both the original VLM training data as well as robotic data for VLM fine-tuning. Since we are mostly interested in the generalization aspects of these models, we remove the *seen tasks* evaluation from this set of experiments.

The results of the ablations are presented in Figure 6b and Appendix Table 7. First, we observe that training a very large model from scratch results in a very poor performance even for the 5B model. Given this result, we decide to skip the evaluation of an even bigger 55B PaLI-X model when trained from scratch. Second, we notice that co-fine-tuning a model (regardless of its size) results in a better generalization performance than simply fine-tuning it with robotic data. We attribute this to the fact that keeping the original data around the fine-tuning part of training, allows the model to not forget its previous concepts learned during the VLM training. Lastly, somewhat unsurprisingly, we notice that the increased size of the model results in a better generalization performance.

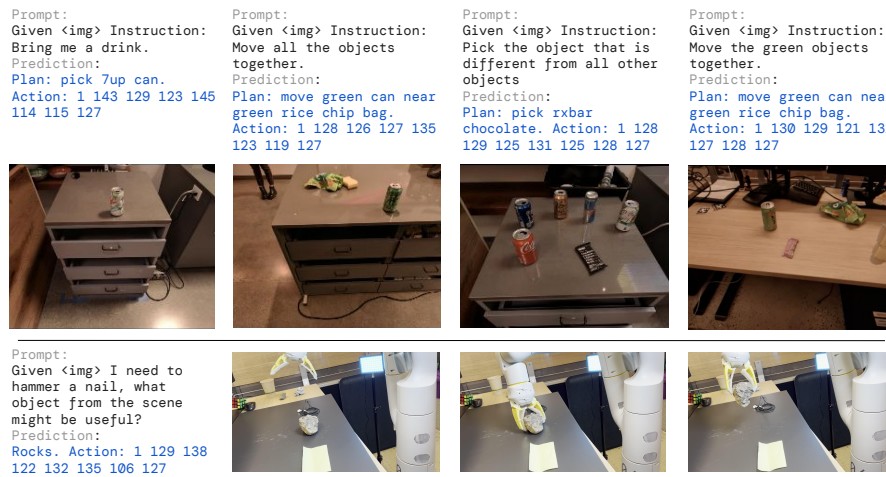

Figure 7: Rollouts of RT-2 with chain-of-thought reasoning, where RT-2 generates both a plan and an action.

### 4.4 Can RT-2 exhibit signs of chain-of-thought reasoning similarly to vision-language models?

Inspired by the chain-of-thought prompting method in LLMs [81], we fine-tune a variant of RT-2 with PaLM-E for just a few hundred gradient steps to increase its capability of utilizing language and actions jointly with the hope that it will elicit a more sophisticated reasoning behavior. We augment the data to include an additional "Plan" step, which describes the purpose of the action that the robot is about to take in natural language first, which is then followed by the actual action tokens, e.g. "Instruction: I'm hungry. Plan: pick rxbar chocolate. Action: 1 128 124 136 121 158 111 255." This data augmentation scheme acts as a bridge between VQA datasets (visual reasoning) and manipulation datasets (generating actions).

We qualitatively observe that RT-2 with chain-of-thought reasoning is able to answer more sophisticated commands due to the fact that it is given a place to plan its actions in natural language first. This is a promising direction that provides some initial evidence that using LLMs or VLMs as planners [19, 17] can be combined with low-level policies in a single VLA model. Rollouts of RT-2 with chain-of-thought reasoning are shown in Figure 7 and in Appendix I.

## 5 Limitations

Even though RT-2 exhibits promising generalization properties, there are multiple limitations of this approach. First, although we show that including web-scale pretraining via VLMs boosts generalization over semantic and visual concepts, the robot does not acquire any ability to perform new *motions* by virtue of including this additional experience. The model's physical skills are still limited to the distribution of skills seen in the robot data (see Appendix G), but it learns to deploy those skills in new ways. We believe this is a result of the dataset not being varied enough along the axes of skills. An exciting direction for future work is to study how new skills could be acquired through new data collection paradigms such as videos of humans.

Second, although we showed we could run large VLA models in real time, the computation cost of these models is high, and as these methods are applied to settings that demand high-frequency control, real-time inference may become a major bottleneck. An exciting direction for future research is to explore quantization and distillation techniques that might enable such models to run at higher rates or on lower-cost hardware. This is also connected to another current limitation in that there are only a small number of generally available VLM models that can be used to create RT-2. We hope that more open-sourced models will become available (e.g. https://llava-vl.github.io/) and the proprietary ones will open up their fine-tuning APIs, which is a sufficient requirement to build VLA models.

## 6 Conclusions

In this paper, we described how vision-language-action (VLA) models could be trained by combining vision-language model (VLM) pretraining with robotic data. We then presented two instantiations of VLAs based on PaLM-E and PaLI-X, which we call RT-2-PaLM-E and RT-2-PaLI-X. These models are co-fine-tuned with robotic trajectory data to output robot actions, which are represented as text tokens. We showed that our approach results in very performant robotic policies and, more importantly, leads to a significantly better generalization performance and emergent capabilities inherited from web-scale vision-language pretraining. We believe that this simple and general approach shows a promise of robotics directly benefiting from better vision-language models, which puts the field of robot learning in a strategic position to further improve with advancements in other fields.

## Acknowledgments

We would like to acknowledge Fred Alcober, Jodi Lynn Andres, Carolina Parada, Joseph Dabis, Rochelle Dela Cruz, Jessica Gomez, Gavin Gonzalez, John Guilyard, Tomas Jackson, Jie Tan, Scott Lehrer, Dee M, Utsav Malla, Sarah Nguyen, Jane Park, Emily Perez, Elio Prado, Jornell Quiambao, Clayton Tan, Jodexty Therlonge, Eleanor Tomlinson, Wenxuan Zhou, Boyuan Chen, and the greater Google DeepMind team for their feedback and contributions.

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

# A Contributions

- **Training and Evaluations (designing and executing procedures for training models, evaluating models in simulation and the real world, running ablations for algorithm design choices)**: Yevgen Chebotar, Krzysztof Choromanski, Tianli Ding, Danny Driess, Avinava Dubey, Pete Florence, Chuyuan Fu, Montse Gonzalez Arenas, Keerthana Gopalakrishnan, Kehang Han, Alexander Herzog, Brian Ichter, Alex Irpan, Isabel Leal, Lisa Lee, Yao Lu, Henryk Michalewski, Igor Mordatch, Karl Pertsch, Michael Ryoo, Anikait Singh, Quan Vuong, Ayzaan Wahid, Paul Wohlhart, Fei Xia, Ted Xiao, and Tianhe Yu.

- **Network Architecture (designing and implementing model network modules, working on tokenization of actions, enabling inference of the model networks during experiments)**: Yevgen Chebotar, Xi Chen, Krzysztof Choromanski, Danny Driess, Pete Florence, Keerthana Gopalakrishnan, Kehang Han, Karol Hausman, Brian Ichter, Alex Irpan, Isabel Leal, Lisa Lee, Henryk Michalewski, Igor Mordatch, Kanishka Rao, Michael Ryoo, Anikait Singh, Quan Vuong, Ayzaan Wahid, Jialin Wu, Fei Xia, Ted Xiao, and Tianhe Yu.

- **Data Collection (collecting data on real robots, running real robot evaluations, executing operations required for running real robots)**: Noah Brown, Justice Carbajal, Tianli Ding, Krista Reymann, Grecia Salazar, Pierre Sermanet, Jaspiar Singh, Huong Tran, Stefan Welker, and Sichun Xu.

- **Leadership (leading the project efforts, managing the project staff, advising on project directions)**: Yevgen Chebotar, Chelsea Finn, Karol Hausman, Brian Ichter, Sergey Levine, Yao Lu, Igor Mordatch, Kanishka Rao, Pannag Sanketi, Radu Soricut, Vincent Vanhoucke, and Tianhe Yu.

- **Paper (working on the paper manuscript, designing paper visualizations and figures)**: Yevgen Chebotar, Danny Driess, Chelsea Finn, Pete Florence, Karol Hausman, Brian Ichter, Lisa Lee, Sergey Levine, Igor Mordatch, Karl Pertsch, Quan Vuong, Fei Xia, Ted Xiao, and Tianhe Yu.

- **Infrastructure (working on infrastructure and code base backbone needed for training models, running experiments, storing and accessing data)**: Anthony Brohan, Yevgen Chebotar, Danny Driess, Kehang Han, Jasmine Hsu, Brian Ichter, Alex Irpan, Nikhil Joshi, Ryan Julian, Dmitry Kalashnikov, Yuheng Kuang, Isabel Leal, Lisa Lee, Tsang-Wei Edward Lee, Yao Lu, Igor Mordatch, Quan Vuong, Ayzaan Wahid, Fei Xia, Ted Xiao, Peng Xu, and Tianhe Yu.

# B Datasets

The vision-language datasets are based on the dataset mixtures from Chen et al. [15] and Driess et al. [17]. The bulk of this data consists of the WebLI dataset, which is around 10B image-text pairs across 109 languages, filtered to the top 10% scoring cross-modal similarity examples to give 1B training examples. Many other captioning and vision question answering datasets are included as well, and more info on the dataset mixtures can be found in Chen et al. [15] for RT-2-PaLI-X, and Driess et al. [17] for RT-2-PaLM-E. When co-fine-tuning RT-2-PaLI-X, we do not use the Episodic WebLI dataset described by Chen et al. [16].

The robotics dataset is based on the dataset from Brohan et al. [1]. This consists of demonstration episodes collected with a mobile manipulation robot. Each demonstration is annotated with a natural language instruction from one of seven skills: "Pick `Object`", "Move `Object` Near `Object`", "Place `Object` Upright", "Knock `Object` Over", "Open `Drawer`", "Close `Drawer`", "Place `Object` into `Receptacle`", and "Pick `Object` from `Receptacle` and place on the counter". Further details can be found in Brohan et al. [1].

RT-2-PaLI-X weights the robotics dataset such that it makes up about 50% of the training mixture for co-fine-tuning. RT-2-PaLM-E weights the robotics dataset to be about 66% of the training mixture.

For the results on Language-Table in Table 1, our model is trained on the Language-Table datasets from Lynch et al. [79]. Our model is co-fine-tuned on several prediction tasks: (1) predict the action, given two consecutive image frames and a text instruction; (2) predict the instruction, given image frames; (3) predict the robot arm position, given image frames; (4) predict the number of timesteps between given image frames; and (5) predict whether the task was successful, given image frames and the instruction.

## C  Baselines

We compare our method to multiple state-of-the-art baselines that challenge different aspects of our method. All of the baselines use the exact same robotic data.

- **RT-1**: Robotics Transformer 1 [1] is a transformer-based model that achieved state-of-the-art performance on a similar suite of tasks when it was published. The model does not use VLM-based pre-training so it provides an important data point demonstrating whether VLM-based pre-training matters.

- **VC-1**: VC-1 [78] is a visual foundation model that uses pre-trained visual representations specifically designed for robotics tasks. We use pre-trained representations from the VC-1 ViT-L model. Since VC-1 does not include language conditioning, we add this by separately embedding the language command via Universal Sentence Encoder [82] to enable comparison to our method. In particular, we concatenate the resulting language embedding tokens to the image tokens produced by VC-1, and pass the concatenated token sequences through token learner [83]. The token sequences produced by token learner are then consumed by an RT-1 decoder-only transformer model to predict robot action tokens. We train the VC-1 baseline end-to-end and unfreeze the VC-1 weights during training, since this led to far better results than using frozen VC-1 weights.

- **R3M**: R3M [58] is a similar method to VC-1 in that R3M uses pre-trained visual-language representations to improve policy training. In this case the authors use Ego4D dataset [84] of human activities to learn the representation that is used by the policy. Both VC-1 and R3M test different state-of-the-art representation learning methods as an alternative to using a VLM. To obtain a language-conditioned policy from the R3M pretrained representation, we follow the same procedure as described above for VC-1, except we use the R3M ResNet50 model to obtain the image tokens, and unfreeze it during training.

- **MOO**: MOO [49] is an object-centric approach, where a VLM is first used to specify the object of interest in a form of a single, colored pixel in the original image. This pixel-modified image is then trained with an end-to-end policy to accomplish a set of manipulation tasks. This baseline corresponds to a situation where a VLM is used as a separate module that enhances perception but its representations are not used for policy learning.

## D  VLMs for RT-2

The PaLI-X model architecture consists of a ViT-22B [85] to process images, which can accept sequences of $n$ images, leading to $n \times k$ tokens per image, where $k$ is the number of patches per image. The image tokens passing over a projection layer is then consumed by an encoder-decoder backbone of 32B parameters and 50 layers, similar to UL2 [86], which jointly processes text and images as embeddings to generate output tokens in an auto-regressive manner. The text input usually consists of the type of task and any additional context (e.g., "Generate caption in ⟨lang⟩" for captioning tasks or "Answer in ⟨lang⟩: question" for VQA tasks).

The PaLI-3B model trained on Language-Table (Table 1) uses a smaller ViT-G/14 [87] (2B parameters) to process images, and UL2-3B [86] for the encoder-decoder network.

The PaLM-E model is based on a decoder-only LLM that projects robot data such as images and text into the language token space and outputs text such as high-level plans. In the case of the used PaLM-E-12B, the visual model used to project images to the language embedding space is a ViT-4B [15]. The concatenation of continuous variables to textual input allows PaLM-E to be fully multimodal, accepting a wide variety of inputs such as multiple sensor modalities, object-centric representations, scene representations and object entity referrals.

## E  Training Details

We perform co-fine-tuning on pre-trained models from the PaLI-X [16] 5B & 55B model, PaLI [15] 3B model and the PaLM-E [17] 12B model. For RT-2-PaLI-X-55B, we use learning rate 1e-3 and batch size 2048 and co-fine-tune the model for 80K gradient steps whereas for RT-2-PaLI-X-5B, we use the same learning rate and batch size and co-fine-tune the model for 270K gradient steps. For RT-2-PaLM-E-12B, we use learning rate 4e-4 and batch size 512 to co-fine-tune the model for 1M gradient steps. Both models are trained with the next token prediction objective, which corresponds to the behavior cloning loss in robot learning. For RT-2-PaLI-3B model used for Language-Table results in Table 1, we use learning rate 1e-3 and batch size 128 to co-fine-tune the model for 300K gradient steps.

# F Evaluation Details

## F.1 Evaluation Scenarios

For studying the emergent capabilities of RT-2 in a quantitative manner, we study various challenging semantic evaluation scenarios that aim to measure capabilities such as reasoning, symbol understanding, and human recognition. A visual overview of a subset of these scenes is provided in Figure 8, and the full list of instructions used for quantiative evalution is shown in Table 4.

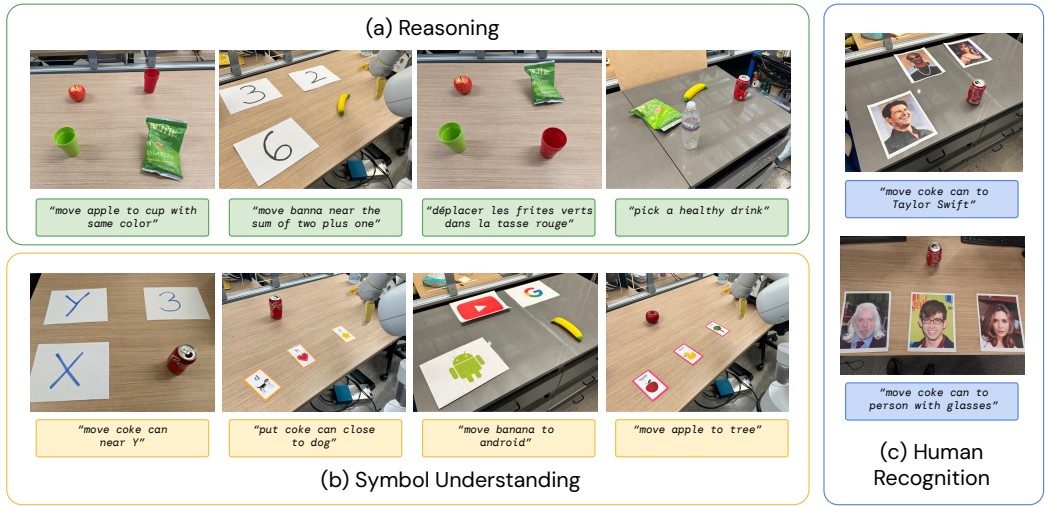

Figure 8: An overview of some of the evaluation scenarios used to study the emergent capabilities of RT-2. They focus on three broad categories, which are (a) reasoning, (b) symbol understanding, and (c) human recognition. The visualized instructions are a subset of the full instructions, which are listed in Appendix F.3.

## F.2 Seen Evaluation Instructions

Table 2 lists the seen robot evaluation instructions, used during model training and seen tasks evaluations.

| Skill | Count | Description | Example Instruction |
|---|---|---|---|
| Pick `Object` | 130 | Lift the object off the surface | pick iced tea can |
| Move `Object` Near `Object` | 337 | Move the first object near the second | move pepsi can near rxbar blueberry |
| Place `Object` Upright | 8 | Place an elongated object upright | place water bottle upright |
| Knock `Object` Over | 8 | Knock an elongated object over | knock redbull can over |
| Open `Drawer` | 3 | Open any of the cabinet drawers | open the top drawer |
| Close `Drawer` | 3 | Close any of the cabinet drawers | close the middle drawer |
| Place `Object` into Receptacle | 84 | Place an object into a receptacle | place brown chip bag into white bowl |
| Pick `Object` from Receptacle and Place on the Counter | 162 | Pick an object up from a location and then place it on the counter | pick green jalapeno chip bag from paper bowl and place on counter |
| Total | 735 | | |

Table 2: The list of skills collected and used during training and seen tasks evaluations, along with their descriptions and example instructions.

## F.3 Unseen Evaluation Instructions

Table 3 lists natural language instructions used in model evaluations for unseen objects, backgrounds, and environments. Each instruction was run between 1-5 times, depending on the number of total instructions in that evaluation set. Table 4 lists natural language instructions used to evaluate quantitative emergent evals. Each instruction was run 5 times.

| Task Group | Tasks |
|---|---|
| Unseen Objects (Easy) | pick banana, move banana near coke can, move orange can near banana, pick oreo, move oreo near apple, move redbull can near oreo, pick pear, pick coconut water, move pear near coconut water, move pepsi can near pear |
| Unseen Objects (Hard) | pick cold brew can, pick large orange plate, pick chew toy, pick large tennis ball, pick bird ornament, pick fish toy, pick ginger lemon kombucha, pick egg separator, pick wrist watch, pick green sprite can, pick blue microfiber cloth, pick yellow pear, pick pretzel chip bag, pick disinfectant wipes, pick pineapple hint water, pick green cup, pick pickle snack, pick small blue plate, pick small orange rolling pin, pick octopus toy, pick catnip toy |
| Unseen Backgrounds (Easy) | pick green jalapeno chip bag, pick orange can, pick pepsi can, pick 7up can, pick apple, pick blue chip bag, pick orange, pick 7up can, move orange near sink, pick coke can, pick sponge, pick rxbar blueberry |
| Unseen Backgrounds (Hard) | pick wrist watch, pick egg separator, pick green sprite can, pick blue microfiber cloth, pick yellow pear, pick pretzel chip bag, pick disinfectant wipes, pick pineapple hint water, pick green cup, pick pickle snack, pick small blue plate, pick small orange rolling pin, pick octopus toy, pick catnip toy, pick swedish fish bag, pick large green rolling pin, pick black sunglasses |
| Unseen Environments (Easy) | pick coke can, pick apple, pick rxbar blueberry, move apple near coke can, move rxbar blueberry near apple, move coke can near rxbar blueberry, pick blue plastic bottle, pick sponge, pick blue chip bag, move sponge near blue plastic bottle, move blue chip bag near sponge, move blue plastic bottle near blue chip bag, move coke can near white mug, move sponge near white mug, move coke can near yellow bowl, move sponge near yellow bowl, move coke can near green cloth, move sponge near green cloth, move coke can near plate, move sponge near plate, move coke can near spoon, move sponge near spoon, move coke can near orange cup, move sponge near orange cup, pick white mug, pick yellow bowl, pick green cloth, move white mug near sponge, move yellow bowl near sponge, move green cloth near sponge, pick plate, pick spoon, pick orange cup, move plate near sponge, move spoon near sponge, move orange cup near sponge, put coke can into sink, drop coke can into sink, push coke can into sink, put sponge into sink, drop sponge into sink, push sponge into sink, put green cloth into sink, drop green cloth into sink, push green cloth into sink |
| Unseen Environments (Hard) | pick coke can, pick apple, pick rxbar blueberry, move apple near coke can, move rxbar blueberry near apple, move coke can near rxbar blueberry, move coke can near stapler, move apple near stapler, move coke can near keyboard, move apple near keyboard, move coke can near tissue box, move apple near tissue box, move coke can near papers, move apple near papers, move coke can near mouse, move apple near mouse, move coke can near book, move apple near book, pick stapler, pick mouse, move marker near apple, move stapler near apple, move mouse near apple, push coke can to the left, push coke can to the right, push sponge to the left, push sponge to the right, push tissue box to the left, push tissue box to the right, point at coke can, point at sponge, point at tissue box |

Table 3: Natural language instructions used for evaluations testing controlled distribution shifts along the dimension of novel objects, novel environments, and novel backgrounds. For each category, we introduce evaluation settings with smaller distribution shifts as well as larger distribution shifts. A visualization of these scenarios if shown in Figure 3.

| Task Group | Tasks |
|---|---|
| Symbol Understanding: Symbol 1 | move coke can near X, move coke can near 3, move coke can near Y |
| Symbol Understanding: Symbol 2 | move apple to tree, move apple to duck, move apple to apple, move apple to matching card |
| Symbol Understanding: Symbol 3 | put coke can close to dog, push coke can on top of heart, place coke can above star |
| Reasoning: Math | move banana to 2, move banna near the sum of two plus one, move banana near the answer of three times two, move banana near the smallest number |
| Reasoning: Logos | move cup to google, move cup to android, move cup to youtube, move cup to a search engine, move cup to a phone |
| Reasoning: Nutrition | get me a healthy snack, pick a healthy drink, pick up a sweet drink, move the healthy snack to the healthy drink, pick up a salty snack |
| Reasoning: Color and Multilingual | move apple to cup with same color, move apple to cup with different color, move green chips to matching color cup, move apple to vaso verde, Bewegen Sie den Apfel in die rote Tasse, move green chips to vaso rojo, mueve la manzana al vaso verde, déplacer les frites verts dans la tasse rouge |
| Person Recognition: Celebrities | move coke can to taylor swift, move coke can to tom cruise, move coke can to snoop dog |
| Person Recognition: CelebA | move coke can to person with glasses, move coke can to the man with white hair, move coke can to the brunette lady |

Table 4: Natural language instructions used for quantitative emergent evalutions.

# G   Example Failure Cases

In Fig. 9 we provide examples of a notable type of failure case in the Language Table setting, with the RT-2 model not generalizing to *unseen object dynamics*. In these cases, although the model is able to correctly attend to the language instruction and move to the first correct object, it is not able to control the challenging

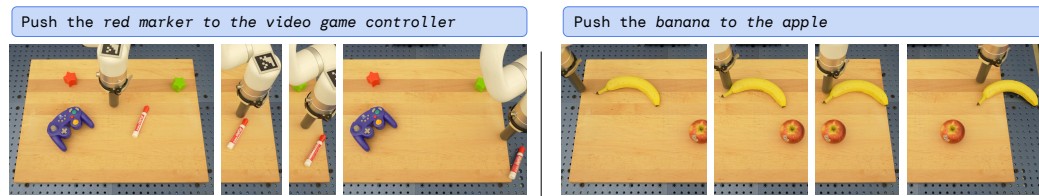

Figure 9: Qualitative example failure cases in the real-world failing to generalize to *unseen object dynamics*.

dynamics of these objects, which are significantly different than the small set of block objects that have been seen in this environment [79]. Then pen simply rolls off the table (Fig. 9, left), while the banana's center-of-mass is far from where the robot makes contact (Fig. 9, right). We note that pushing dynamics are notoriously difficult to predict and control [88]. We hypothesize that greater generalization in robot-environment interaction dynamics may be possible by further scaling the datasets across diverse environments and objects – for example, in this case, datasets that include similar types of more diverse pushing dynamics [33].

In addition, despite RT-2's promising performance on real world manipulation tasks in qualitative and quantitative emergent evaluations, we still find numerous notable failure cases. For example, with the current training dataset composition and training method, RT-2 seemed to perform poorly at:

- Grasping objects by specific parts, such as the handle
- Novel motions beyond what was seen in the robot data, such as wiping with a towel or tool use
- Dexterous or precise motions, such as folding a towel
- Extended reasoning requiring multiple layers of indirection

# H Quantitative Experimental Results

## H.1 Overall Performance, for Section 4.1

Table 5 lists our quantitative overall evaluation results. We find that RT-2 performs as well or better than baselines on seen tasks and significantly outperforms baselines on generalization to unseen objects, backgrounds, and environments.

| Model | Seen Tasks | Unseen Objects | | Unseen Backgrounds | | Unseen Environments | | Unseen Average |
|---|---|---|---|---|---|---|---|---|
| | | Easy | Hard | Easy | Hard | Easy | Hard | |
| R3M [58] | 45 | 32 | 14 | 13 | 9 | 0 | 2 | 12 |
| VC-1 [78] | 63 | 34 | 10 | 13 | 3 | 0 | 0 | 10 |
| RT-1 [1] | **92** | 31 | 43 | 71 | 9 | 26 | 14 | 32 |
| MOO [49] | 75 | 58 | 48 | 38 | 41 | 19 | 3 | 35 |
| RT-2-PaLI-X-55B (ours) | 91 | 70 | 62 | 96 | 48 | 63 | 35 | 62 |
| RT-2-PaLM-E-12B[2] (ours) | 93 | 84 | 76 | 75 | 71 | 36 | 33 | 62 |

Table 5: Overall performance of two instantiations of RT-2 and baselines across seen training tasks as well as unseen evaluations measuring generalization to novel objects, novel backgrounds, and novel environments.

## H.2 Emergent Evaluation, for Section 4.2

Table 6 lists all of our quantitative emergent evaluation results. We find that RT-2 performs 2x to 3x better than RT-1 on these new instructions, without any additional robotic demonstrations. This showcases how our method allows us to leverage capabilities from pretraining on web-scale vision-language datasets.

| Model | Symbol Understanding | | | | Reasoning | | | | | Person Recognition | | | Average |
|---|---|---|---|---|---|---|---|---|---|---|---|---|---|
| | Symbol 1 | Symbol 2 | Symbol 3 | Average | Math | Logos | Nutrition | Color/Multilingual | Average | Celebrities | CelebA | Average | |
| VC-1 [78] | 7 | 25 | 0 | 11 | 0 | 8 | 20 | 13 | 10 | 20 | 7 | 13 | 11 |
| RT-1 [1] | 27 | 20 | 0 | 16 | 5 | 0 | 32 | 28 | 16 | 20 | 20 | 20 | 17 |
| RT-2-PaLI-X-55B (ours) | **93** | **60** | **93** | **82** | 25 | 52 | **48** | **58** | **46** | **53** | 53 | **53** | **60** |
| RT-2-PaLM-E-12B (ours) | 67 | 20 | 20 | 36 | **35** | **56** | 44 | 35 | 43 | 33 | **53** | 43 | 40 |

Table 6: Performance of RT-2 and baselines on quantitative emergent evaluations.

---

[2]The original pre-training data mixture used in PaLM-E-12B (as described in [17]) includes robot images for high-level VQA planning tasks that can be similar to images encountered in generalization scenarios. However, none of those training examples include low-level actions that are evaluated in this experiment.

### H.3 Size and Training Ablations, for Section 4.3

Table 7 details quantitative results for ablations across model size and training approach. Across each, we see that model size plays an important role in performance and that co-fine-tuning outperforms fine-tuning, which outperforms training from scratch.

| Model | Size | Training | Unseen Objects | | Unseen Backgrounds | | Unseen Environments | | Average |
|---|---|---|---|---|---|---|---|---|---|
| | | | Easy | Hard | Easy | Hard | Easy | Hard | |
| RT-2-PaLI-X | 5B | from scratch | 0 | 10 | 46 | 0 | 0 | 0 | 9 |
| RT-2-PaLI-X | 5B | fine-tuning | 24 | 38 | 79 | 50 | 36 | 23 | 42 |
| RT-2-PaLI-X | 5B | co-fine-tuning | 60 | 38 | 67 | 29 | 44 | 24 | 44 |
| RT-2-PaLI-X | 55B | fine-tuning | 60 | 62 | 75 | 38 | 57 | 19 | 52 |
| RT-2-PaLI-X | 55B | co-fine-tuning | 70 | 62 | 96 | 48 | 63 | 35 | 63 |

Table 7: Ablations of RT-2 showcasing the impact of parameter count and training strategy on generalization.

## I   Additional Chain-Of-Thought Reasoning Results

We present additional examples of chain-of-thought reasoning rollouts accomplished with RT-2-PaLM-E, as described in Sec. 4.4, in Figure 10.

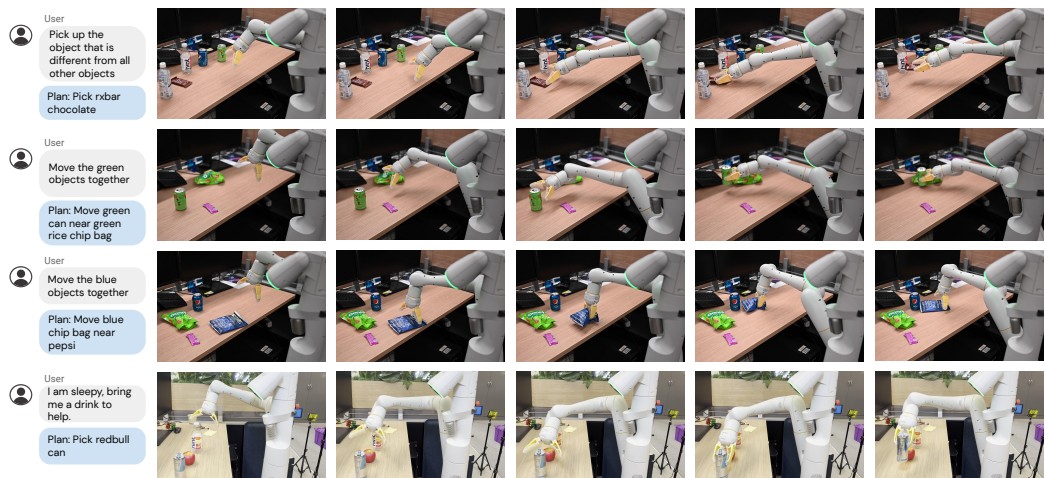

Figure 10: Additional examples of RT-2 with chain-of-thought reasoning

