# OpenReview forum: "RT-2: Vision-Language-Action Models Transfer Web Knowledge to Robotic Control"
_robot-learning.org/CoRL/2023/Conference — CoRL 2023 Poster_

### Official Review · Reviewer_N1nv · 2023-07-19

**Confidence:** 5
**Originality:** Very Good
**Technical Quality:** Very Good
**Clarity Of Presentation:** Very Good
**Impact:** 3

**Recommendation:**

Weak Accept: I recommend accepting the paper, but will not argue for my recommendation if the majority of other reviewers have a different opinion.

**Review:**

The paper presents an interesting concept, Vision Language Action models: up to now, Large Language Models (LLMs) and VLMs had been used in robotics in conjunction with other models. They were mostly adopted as high-level planners, querying often object detection models, and also skill modules that ultimately outputted actions. This paper demonstrates that a single VLM, finetuned also on action data, can unify the process and take advantage of the knowledge stores in the billions of weights of the VLM, distilled from web-scale datasets. The authors also compare it to existing baselines and study the emergent properties of such models.

However, my concern is the kind of tasks evaluated in this paper, as also highlighted by the authors in the Limitations section: most of the tasks boil down to pick and place tasks. Most of the object, additionally, can be grasped by simply going to the centroid of the object and closing the gripper: for example, in a video on the website, the robot grasps a paper cup by squeezing it instead of grasping it from e.g. the brim. All the emerging capabilities are fundamentally vision-recognition and ability to recognise a target area: while generally still remarkable, this is a know property of VLMs.

Therefore, while I agree with the authors when they say, in the Related Work section, that methods like CLIPort require 2D action spaces, the tasks they presented could mostly be solved with this kind of action space, with CLIPort likely also doing a good job at recognising  target areas. The current emerging capabilities of RT-2 do not really suggest improvement in real, everyday like situations. While the capability to, e.g., solve "put the strawberry in the correct bowl" is interesting, this is still something that could likely be done with existing techniques, such as Code as Policies and CLIPort.

I encourage the authors to focus, likely in future papers, on expanding the skills repertoire of the robot in the demonstrations recording phase: observing emerging abilities in the skills area would really be a breakthrough for the robotics community, and strongly suggest that these models can provide unprecedented capabilities. At the moment, the line of work composed of the papers mentioned here (RT-1, PaLM-E, MOO), while interesting, do not push enough the dexterity side of robotics, needed to interact with human environments.

**Quality Of The Limitations Section:**

Limitations are addressed clearly

**Questions For Rebuttal:**

While I realise that implementing and evaluating a new baseline, like CLIPort, can be too much work in the rebuttal window, I encourage the reviewers to better discuss why their current tasks could not be solved by that method, or similar methods. As I mentioned above, many tasks seem to involve 1) finding an object and grasping it from the centroid 2) finding a target, albeit after receiving a nuanced instruction, and putting the object there.

**Robotics Focus:**

Sufficient demonstration on hardware

**Summary Of Paper:**

This paper introduces Vision Language Action (VLAs) models: in a nutshell, Vision Language Models (VLMs) finetuned to also express robot actions in "language form", or more precisely as tokens, therefore unifying the output space of language and actions. The authors finetune several VLMs on robotics data, in particular demonstration trajectories of various tasks, and co-finetune on "normal" vision tasks for which language is expected as output.

The authors demonstrate that this enables a robot to receive language commands and predict actions directly through a single model, in contrast to other works that generally use a set of models to bridge language, vision, and action. The authors demonstrate that their method, RT-2, while being on par on seen tasks with some other existing models (e.g. RT-1), clearly surpasses them on unseen tasks. They also demonstrate emerging capabilities such as the ability to understand more nuanced commands or the ability to perform more complex reasoning.

**Summary Of Recommendation:**

Given the novelty of the VLAs ideas, the numerous experiments, and the ablations and investigations on the emerging capabilities of these models, I suggest a weak accept. However, as expressed before, I do believe that the method feels like a pick-and-place method able to recognise targets and objects even when instructions are not-direct and nuanced. The rebuttal discussion from the authors, that I hope will better justify why these methods will be able to become much more dexterous, can influence my final decision.

---

### Official Review · Reviewer_YNc7 · 2023-07-20

**Confidence:** 3
**Originality:** Very Good
**Technical Quality:** Good
**Clarity Of Presentation:** Excellent
**Impact:** 4

**Recommendation:**

Weak Accept: I recommend accepting the paper, but will not argue for my recommendation if the majority of other reviewers have a different opinion.

**Review:**

**Originality & Significance**

RT-2 produces state-of-the-art results on a dataset introduced in the RT-1 paper. It outperforms the RT-1 methods on the dataset. Further papers, such as an exploration into multi-step planning, could build on this work.

**Quality & Clarity**

The paper was very well written and clear. Experiments were conducted using carefully designed tasks which measure the model's generalizability to different types of challenges.

**Strengths**
- The results provided in this paper provide exciting results. The paper demonstrates improvement over RT-1 methods.
- The model's reasoning and generalizability are impressive. Inputted language can be flexible. Generalizable across camera, environment, & setup.
- The writing and figures were clear and comprehensive.

**Weaknesses**
- All baselines used for comparison are based on the RT-1 architecture. The paper could be strengthened with the inclusion of other methods of incorporating LLMs. While the setup is slightly different (no direct training of VLA), works such as "Code-As-Policies" [1] seem like a comparison to add.
- Quantitative data on chain-of-thought reasoning would be a plus. Discussion into the level of permissible abstraction in the prompt would also be interesting.
- Although the model is able to adapt to new language, the RT-2 paradigm is only demonstrated on a the narrow application of robot manipulators. Could RT-2 be applied to other tasks (mobile robots, quadrupeds, etc.)

[1] Code as Policies: Language Model Programs for Embodied Control, arXiv:2209.07753


**Quality Of The Limitations Section:**

Limitations are addressed clearly

**Questions For Rebuttal:**

- Please define the distinction between "easy" and "hard" tasks from Table 1? Even looking at appendix Table 3, I am unclear about the difference between the two.
- In the Output Constraints section, can you please elaborate on how you "constrain [the model's] output vocabulary via only sampling valid action tokens when the model is prompted with a robot-action task"? How do you prevent the model from outputting language tokens in this case?

**Robotics Focus:**

Sufficient demonstration on hardware

**Summary Of Paper:**

The paper explores an approach to generating low-level robotic actions from open vocabulary natural language commands. The proposed method finetunes large pretrained visual-language models (VLM) using a robot manipulator dataset consisting of simple atomic actions and instruction annotations. The authors demonstrate the model's promising capabilities to converge on the robot dataset and generalize to unseen objects, backgrounds, and environments when compared to SOTA baselines.

**Summary Of Recommendation:**

While the paper lacked broad comparison to other LLM-integrated zero-shot robotic learning methods, it presents an novel direction and promising results. I recommend accepting the paper to this conference.

---

### Official Review · Reviewer_Bmbe · 2023-07-20

**Confidence:** 4
**Originality:** Very Good
**Technical Quality:** Very Good
**Clarity Of Presentation:** Very Good
**Impact:** 4

**Recommendation:**

Weak Accept: I recommend accepting the paper, but will not argue for my recommendation if the majority of other reviewers have a different opinion.

**Review:**

Strengths:
  1. This paper proposes a novel method to tokenize robot actions into text tokens, which allows robot trajectory data to be directly used to fine-tune VLMs.
  2. The paper is well-motivated and clearly written, and emergent capabilities can be observed from the videos. RT-2 enables the robot to understand ambiguous instructions and complete the corresponding task.

Weaknesses:
  1.  As the authors addressed in Sec 3.3, the computation cost of these models is high. The real-time inference relies on multi-TPU cloud service, which limits the application scenarios.
  2. The evaluation skills are mainly pick-and-place skills, lacking the manipulation of the articulated objects (included in the robotic trajectory data) .
  3. The trajectory of the robot manipulator is observed to be less stable than RT-1 from the video. It seems that control of the end-to-end VLA is less robust.

**Quality Of The Limitations Section:**

Limitations are addressed clearly

**Questions For Rebuttal:**

1. The authors co-fine-tuned separate models for different experimental  setups. I am curious whether robot trajectory data from different robot morphologies can be used to train a single end-to-end VLA.
2. Can RT-2 complete more complex tasks, such as opening a cabinet or closing a drawer?
3. Can RT-2 complete long-horizon / multi-step tasks like Saycan end-to-end?

**Robotics Focus:**

Sufficient demonstration on hardware

**Summary Of Paper:**

This paper introduces RT-2, a method to incorporate end-to-end robotic control into large vision-language models. By representing actions as text tokens, robotic trajectory data can be directly used to fine-tune VLMs. Experiments demonstrate that RT-2 can absorb emergent capabilities for robotic manipulation from Internet-scale training.

**Summary Of Recommendation:**

Overall, this paper proposes a novel idea to directly use robot trajectory data for the training of VLMs. The paper is well-presented and the experimental results are convincing. To further improve the paper, more skills (such as opening and closing) could be included in the experiments.

---

### Official Review · Reviewer_UgBu · 2023-07-21

**Confidence:** 5
**Originality:** Good
**Technical Quality:** Fair
**Clarity Of Presentation:** Very Good
**Impact:** 4

**Recommendation:**

Weak Accept: I recommend accepting the paper, but will not argue for my recommendation if the majority of other reviewers have a different opinion.

**Review:**

This paper provides some of the first concrete validation that VLMs can be used to directly improve the performance of low-level robot behavior. Notably (and the authors acknowledge this) the result primarily demonstrates the usefulness of VLMs in adding semantic understanding to simple visuomotor polices (such as selecting objects by size) and in increasing the robustness of the learned policies to shift in object type, scene/background, and environment setup. They do not find a significant improvement in in-distribution performance of these behaviors compared to learning them directly (the RT-1 comparison).

The biggest shortcoming of this work is a lack of careful ablation. A few of the most valuable ones that were omitted are:

- A comparison of this approach on single tasks with SOTA single-task BC methods such as ALOHA, DiffusionPolicy, or IBC.

- An ablation over amount of robot data and the ratio of robot data to VQA data.

- A comparison of both PALM and PALM-E results which would provide an apples-to-apples comparison of what impact the higher-level embodied task data present in PALM-E had. Gleaning this from the PaLI-X/PALM-E comparison is difficult due to differences between the two models.

- A quantitative comparison between the chain-of-thought and vanilla versions of RT-2.

- A quantitative comparison between PaLI-X/PALM-E and some of the existing open-source VLMs currently available such as BLIP-2 or LLAVA.

Notably, performing many of these ablations on physical hardware may have been prohibitive, but the authors included some results in a simulated tabletop setting where this would have still been valuable.

**Quality Of The Limitations Section:**

Limitations are addressed clearly

**Questions For Rebuttal:**

- Why was the "seen task" category removed from Appendix H? Even if generalization is expected to be the most interesting aspect of this experiment, it's still important to be measuring and reporting on effect (or lack thereof) on in-distribution performance?

- Did chain of thought reasoning notably impact overall performance (both in and out of distribution) of the robot?

**Robotics Focus:**

Sufficient demonstration on hardware

**Summary Of Paper:**

RT-2 finetunes a pretrained VLM (in this case versions of PaLM-E and PaLI-X) on a mixture of web VLM training data (e.g. VQA) and embodied robot data to directly produce robot action commands conditioned on camera observations and a language prompt. This work discusses several relevant design decisions such as cloud serving, mixing VQA data with robot data during finetuning, and selection of output tokens for action (discretize the action space as in RT-1 and either use numerical tokens if appropriate or overwrite least used tokens).

They conduct a large number of manipulation (mainly object pick-and-place and pushing) experiments both in hardware and in simulation and show strong evidence that coopting a pretrained VLM use useful for imparting some of the semantic understanding of a VLM onto the learned visuomotor policies and for reducing sensitivity to distribution and domain shift.

**Summary Of Recommendation:**

This paper clearly provides a valuable contribution to the field. That being said, given the resourcing that clearly went into this work and the scope of the paper there are obvious experiments, comparisons, and details that are omitted that would dramatically improve the rigor and scientific value.

---

### Author Response · Authors · 2023-08-11
**Response to all Reviewers**

We thank the reviewers for all the helpful comments and suggestions. We specify the main changes to the paper below and provide more detailed responses in the respective replies.

We will add a new table to the manuscript to reflect the distribution of tasks that RT-2 is trained on. Please see it below:

| Skill                        | Count  | Description                              |   Example Instruction |
| --- | --- | --- | --- |
| Pick Object                  | 130    | Lift the object off the surface          |   pick iced tea can |
| Move Object Near Object      | 337    | Move the first object near the second    |   move pepsi can near rxbar blueberry |
| Place Object Upright         | 8      | Place an elongated object upright        |   place water bottle upright |
| Knock Object Over            | 8      | Knock an elongated object over           |   knock redbull can over |
| Open Drawer                  | 3      | Open any of the cabinet drawers          |   open the top drawer |
| Close Drawer                 | 3      | Close any of the cabinet drawers         |   close the middle drawer |
| Place Object into Receptacle | 84     | Place an object into a receptacle place  |   brown chip bag into white bowl |
| Pick Object from Receptacle and Place on the Counter | 162    | Pick an object up from a location and then place it on the counter   |   pick green jalapeno chip bag from paper bowl and place on counter|

We acknowledge that the tasks demonstrated here are not physically very complex and we’ll add this discussion to the paper. However, we would also like to clarify that there are a few skills suggested by the reviewers (e.g. opening and closing). In particular, include 18 drawer opening and closing tasks (L217), 35 tasks involving carefully placing objects upright (L216), and object pushing tasks on a table top in the language-table scenario.

To investigate chain-of-thought quantitatively, we ran each instruction (five times) in the emergent reasoning category “Color and Multilingual”. This includes tasks such as “move apple to cup with same color” and “déplacer les frites verts dans la tasse rouge”. We chose this category as the task performed was furthest from the instruction given and thus the chain of thought was more interesting than simply repeating the instruction requested.  Our results showed that PaLM-E-12B without chain of thought achieved 35%, while with chain of thought achieved 42.5% (a 7.5% performance increase). Though this gain is modest, we expect the gains will be further increased as the instructions become more complex and out of distribution, such as several of the tasks in Figure 7.

| Success Rate w/o CoT | Success Rate w/ CoT |
| --- | --- |
| 35% | 42.5% |

We’ll add these results to the manuscript.

---

### Decision · Program_Chairs · 2023-08-30

**Decision:**

Accept (Poster)

**Comment:**

This paper presents a method for fine-tuning a pre-trained VLM simultaneously on robotics data and web-scale vision-language tasks. Real-world experiments show that this enables significant generalisation of language-conditioned robot manipulation policies to new, previously unseen tasks. The trained network shows emerging capabilities, such as understanding abstract or nuanced language commands, as opposed to more straightforward language commands that we have typically seen before in language-conditioned robot learning.

Reviews before the rebuttal were 4 x “weak accept”. Reviewers were positive about the novelty and ambition of this idea, and intrigued by the impressive generalisation ability. Whilst there were no major concerns, reviewers did raise some points about the lack of in-depth ablations, and the limited range of manipulation skills that RT-2 is able to perform (mainly high tolerance tasks, such as pick-and-place).

Authors responded to questions and made some minor changes to the paper, but did not provide any further experiments. However, for a large-scale project of this nature, this is understandable given the short time period available. Therefore, after the rebuttal, all 4 reviewers retained their “weak accept” scores, but none decided to upgrade.

Overall, we find that this paper presents an intriguing and promising future direction for robot learning as these models continue to scale, and the emerging capabilities of RT-2 are certainly very exciting. Many of the reviewers’ comments have already been incorporated into the updated paper. But there are still further ablations and more in-depth investigations that could be incorporated into the final paper if time permits, because the paper’s experiments mainly focus on what the abilities of RT-2 are, rather than providing introspection as to why these abilities emerge. For future work beyond this paper, we also encourage the authors to address more complex manipulation tasks than these high-tolerance pick-and-place tasks.